# PNPLA3-I148M Variant Promotes the Progression of Liver Fibrosis by Inducing Mitochondrial Dysfunction

**DOI:** 10.3390/ijms24119681

**Published:** 2023-06-02

**Authors:** Yusong Gou, Lifei Wang, Jinhan Zhao, Xiaoyi Xu, Hangfei Xu, Fang Xie, Yanjun Wang, Yingmei Feng, Jing Zhang, Yang Zhang

**Affiliations:** 1The Third Unit, The Department of Hepatology, Beijing Youan Hospital, Capital Medical University, Beijing 100069, China; yusonggou@ccmu.edu.cn (Y.G.); zhaojinhan@mail.ccmu.edu.cn (J.Z.); xiaoyixu2020@163.com (X.X.); xuhangfei@mail.ccmu.edu.cn (H.X.); 2Beijing Institute of Hepatology, Beijing Youan Hospital, Capital Medical University, Beijing 100069, China; xfxf815@126.com (F.X.); yjunwang@ccmu.edu.cn (Y.W.); 3NHC Key Laboratory of Biotechnology of Antibiotics, Institute of Medicinal Biotechnology, Chinese Academy of Medical Sciences & Peking Union Medical College, Beijing 100050, China; lifeiwang2002@hotmail.com; 4Beijing Engineering Research Center for Precision Medicine and Transformation of Hepatitis and Liver Cancer, Beijing 100069, China; 5Beijing Youan Hospital, Capital Medical University, Beijing 100069, China; yingmeif13@sina.com

**Keywords:** PNPLA3-I148M, hepatic stellate cells, liver fibrosis, free cholesterol, mitochondrial

## Abstract

Patatin-like phospholipase domain-containing 3 (PNPLA3) rs738409 polymorphism (I148M) is strongly associated with non-alcoholic steatohepatitis and advanced fibrosis; however, the underlying mechanisms remain largely unknown. In this study, we investigated the effect of PNPLA3-I148M on the activation of hepatic stellate cell line LX-2 and the progression of liver fibrosis. Immunofluorescence staining and enzyme-linked immunosorbent assay were used to detect lipid accumulation. The expression levels of fibrosis, cholesterol metabolism, and mitochondria-related markers were measured via real-time PCR or western blotting. Electron microscopy was applied to analyze the ultrastructure of the mitochondria. Mitochondrial respiration was measured by a Seahorse XFe96 analyzer. PNPLA3-I148M significantly promoted intracellular free cholesterol aggregation in LX-2 cells by decreasing cholesterol efflux protein (ABCG1) expression; it subsequently induced mitochondrial dysfunction characterized by attenuated ATP production and mitochondrial membrane potential, elevated ROS levels, caused mitochondrial structural damage, altered the oxygen consumption rate, and decreased the expression of mitochondrial-function-related proteins. Our results demonstrated for the first time that PNPLA3-I148M causes mitochondrial dysfunction of LX-2 cells through the accumulation of free cholesterol, thereby promoting the activation of LX-2 cells and the development of liver fibrosis.

## 1. Introduction

Nonalcoholic fatty liver disease (NAFLD) is a prevalent disease initially characterized by the excessive accumulation of lipids in the liver, which can progress to nonalcoholic steatohepatitis (NASH) and fibrosis, subsequently leading to an increased risk of cirrhosis and hepatocellular carcinoma (HCC) [1,2,3]. As a key factor of NAFLD, liver fibrosis is a common histological change of liver injury due to the excessive accumulation of extracellular matrix (ECM) proteins [4,5]. Hepatic stellate cells (HSCs), located in the perisinusoidal space, have been demonstrated to be involved in the pathogenesis of liver fibrosis as major collagen- and ECM-producing cells [6]. In the normal liver, HSCs remain in a quiescent state and serve as the primary storage cells for vitamin A [7,8]. In response to pathological stimuli, HSCs can migrate to the site of injury and are further activated by transdifferentiation into myofibroblast-like cells with a proliferative, migratory, and fibrotic phenotype [9]. The activation of HSCs is thought to be a key cellular event in hepatic fibro-genesis because of their ability to secrete excess extracellular matrix proteins, such as type I collagen, α-smooth muscle actin, and pro-inflammatory cytokines, leading to the accumulation of a scar matrix and contributing to liver injury [9,10].

Several pathways or mediators, such as autophagy, endoplasmic reticulum stress, oxidative stress, retinol, inflammatory cytokines, free cholesterol (FC), reactive oxygen species (ROS), etc., are involved in the initiation of HSC activation [11]. Among them, FC accumulation has been reported to play a key role in the pathogenesis of liver fibrosis. Tomita, et al. [12] found that the increase in cholesterol intake promotes liver fibrosis in mice mainly through the accumulation of FC in HSCs but not by affecting hepatocyte damage or Kupffer cell activation. Furthermore, imbalances in the cholesterol metabolism in HSCs can promote oxidative stress and altered mitochondrial dynamics, which in turn leads to mitochondrial dysfunction and the susceptibility of HSCs to apoptosis [13,14]. Therefore, the excessive aggregation of FC in HSCs is also considered an independent risk factor for the development of liver fibrosis.

Patatin-like phospholipase domain containing 3 (PNPLA3) belongs to the PNPLA3 family of proteins with triacylglycerol lipase and acylglycerol O-acyltransferase activities and is localized on the surface of lipid drops (LDs) [15,16,17]. The common non-synonymous gene variant (C > G, rs738409) of PNPLA3, which substitutes isoleucine with methionine at position 148 (PNPLA3-I148M), has been strongly associated with a susceptibility to NAFLD [15,18]. Mechanistic studies have revealed that PNPLA3-I148M can evade ubiquitinated degradation under starvation conditions in vivo, inhibiting adipose triglyceride lipase (ATGL) activity by sequestering the ATGL cofactor α/β-hydrolase domain-containing protein 5 (ABHD5) and subsequently leading to lipid accumulation in hepatocytes [19]. PNPLA3-I148M has also been reported to cause liver fibrosis by reducing its retinyl-palmitate hydrolytic activity, promoting intercellular matrix metalloproteinase expression and fibrotic remodeling in HSCs [20,21,22]. Moreover, PNPLA3-I148M was also shown to cause cholesterol accumulation in the immortalized human stellate cell line (LX-2) and mouse liver cells, respectively, suggesting that it has an effect on cholesterol metabolism [23,24]. However, the underlying mechanism of PNPLA3-I148M on fibrosis and the imbalance of cholesterol metabolism remains unknown.

In this study, we evaluated the effects of PNPLA3-I148M on cholesterol metabolism and mitochondrial function in LX-2. The results indicate that PNPLA3-I148M induced intracellular FC accumulation, which further contributed to the dysfunction of the mitochondria and the progressive fibrotic phenotype of HSCs. Thus, this study provides a new possible mechanism by which mitochondrial dysfunction is a key intermediate within PNPLA3-I148M-driven HSC activation.

## 2. Results

### 2.1. PNPLA3-I148M Promotes Intracellular LD Accumulation and LX-2 Activation

We first identified the expression of PNPLA3 in LX-2 cells. Both the mRNA and protein levels of PNPLA3-I148I or -I148M were significantly increased in stable cell lines compared to the EV control (*p* < 0.001) (Figure 1A–C). As for the accumulation of LDs, incubation with PA-retinol or oleic acid (OA)-retinol significantly increased intracellular LD levels in all three groups of LX-2 cells (Figure 1D, Appendix A). However, the number of LDs in PNPLA3-I148M was twice that of the PNPLA3-I148I and EV cells (*p* < 0.01) (Figure 1D,F). After PA-retinol or OA-retinol depletion, the PNPLA3-I148M cells exhibited a sustained high level of lipid droplets (*p* < 0.001 vs. EV or PNPLA3-I148I), while the PNPLA3-I148I cells showed a significant reduction (*p* < 0.05 vs. EV) (Figure 1E,G, Appendix A). Co-localization studies also revealed that both PNPLA3-I148I and -I148M proteins localized on the membrane surface of LDs under different stimulation conditions (PA-retinol or OA-retinol) (Appendix A).

Furthermore, PNPLA3-I148M cells also exhibited an increase in the gene expression levels of *acetyl-CoA carboxylase α-1 (ACC-1)* and *fatty acid synthase (FAS)* (*p* < 0.05), which are involved in fatty acid synthesis. These levels remained significantly high, even after the removal of PA-retinol (*p* < 0.05 for *ACC-1*; *p* < 0.001 for *FAS* vs. PNPLA3-I148I) (Appendix A). A reduced expression of the *adipose triglyceride lipase (ATGL)*, *hormone-sensitive lipase (HSL)*, and *monoacylglycerol lipase (MGL)* genes was found in PNPLA3-I148M cells compared to PNPLA3-I148I; however, neither ATGL nor CGI-58 showed differences in protein expression levels under these two stimulation conditions (Appendix A).

For LX-2 cell activation, we found that PNPLA3-I148M cells exhibited high expression levels of *COL3A1* (*p* < 0.01), *TIMP1* (*p* < 0.01), and *TIMP2* (*p* < 0.05 for incubation condition; *p* < 0.001 for depletion condition) genes under both stimulation conditions (Appendix A). However, *COL1A1* (*p* < 0.001), *MMP2* (*p* < 0.001), and *MLC-2* (*p* < 0.05) gene expression was significantly increased only under PA-retinol depletion conditions compared to PMPLA3-I148I (Appendix A). This suggests that the removal of PA-retinol exacerbates the development of the PNPLA3-I148M-induced fibrotic phenotype in LX-2 cells.

The above results suggest that PNPLA3-I148M promotes lipid droplet aggregation, lipogenesis, and fibrogenesis in LX-2 cells, which is consistent with other studies [20].

### 2.2. PNPLA3-I148M Induces FC Accumulation in LX-2 Cells

To assess the effects of PNPLA3-I148M on lipid homeostasis, we tested the levels of TG, TC, and FC in these stable LX-2 cell lines. Compared with PNPLA3-I148I, incubation with PA-retinol enhanced the levels of supernatant TC (*p* < 0.05), while PA-retinol removal markedly increased the supernatant FC (*p* < 0.05) (Figure 1H–M). Importantly, we found that the intracellular TG (*p* < 0.001), TC (*p* < 0.01 for incubation condition; *p* < 0.001 for depletion condition), and FC (*p* < 0.05 for incubation condition; *p* < 0.001 for depletion condition) levels in PNPLA3-I148M cells were significantly higher than in PNPLA3-I148I cells, regardless of PA removal or incubation conditions (Figure 1I,K,M). The observed increase in TC and FC levels suggests that PNPLA3-I148M has a significant impact on the pathways involved in cholesterol metabolism in LX-2 cells.

We conducted additional analysis to investigate the accumulation of FC in LX-2 cells using filipin staining. The results showed that the levels of FC in PNPLA3-I148M cells were significantly higher than that in EV (*p* < 0.05) or PNPLA3 I148I (*p* < 0.01 for incubation condition; *p* < 0.001 for depletion condition) (Figure 2A–C), which is consistent with our previous observations (Figure 1M).

For the levels of proteins and genes involved in cholesterol metabolism, PNPLA3-I148M cells showed a significant increase in the expression of ATP-binding cassette subfamily A member 1 (ABCA1), a protein involved in cholesterol efflux, compared to PNPLA3-I148M under both stimulation conditions (Figure 2D). In addition, significantly reduced ATP-binding cassette subfamily G member 1 (ABCG1) (*p* < 0.05) and elevated Acyl CoA: Cholesterol O-acyl transferases (ACAT) (*p* < 0.001) proteins were also found in PNPLA3-I148M cells only under depletion condition; however, endogenous cholesterol synthesis protein-sterol-regulated HMG-CoA reductase (HMGCR), cholesterol uptake protein-low density lipoprotein receptor (LDLR), and sterol regulatory element binding transcription factor 2 (SREBP2) did not show the differences (Figure 2D). The gene expression levels of the above proteins were also significantly altered under different stimulation conditions, especially after PA removal (Appendix A).

### 2.3. PNPLA3-I148M Disrupts the Function of Mitochondrial

Increasing evidence has supported the notion that FC load is closely related to mitochondrial dysfunction in NASH [25,26]. On this basis, we further evaluated mitochondrial function in PNPLA3-I148I and -I148M stably expressed LX2 cells. PA-retinol incubation significantly reduced the intracellular ATP levels in PNPLA3-I148M cells when compared to PNPLA3-I148I (*p* < 0.001) (Figure 3A,B). Both supernatant and intracellular ATP levels, as well as ΔΨm, were significantly lower in PNPLA3-I148M compared with PNPLA3-I148I cells after PA-retinol depletion (*p* < 0.001 for ATP; *p* < 0.001 for ΔΨm) (Figure 3A–C). In addition, we further found that PNPLA3- I148M cells had significantly high levels of reactive oxygen species (ROS) under both conditions compared to PNPLA3-I148I (*p* < 0.05 for incubation; *p* < 0.001 for depletion) (Figure 3D). However, reduced ATP levels, ΔΨm, and increased ROS reflected the effects of PNPLA3-I148M on mitochondrial dysfunction in LX-2 cells.

### 2.4. Impact of PNPLA3-I148M on Mitochondrial Ultrastructure

To further detect the impact of PNPLA3-I148M on mitochondria, we observed the morphology of the mitochondria of LX-2 stable cell lines by a transmission electron microscope. Both EV and PNPLA3-148I LX-2 cells displayed moderate numbers of mitochondria with an intact outer membrane and slightly disorganized cristae arrangement (Figure 3E,F). Contrastingly, PNPLA3-I148M LX-2 cells had an increased number of mitochondria, increased outer membrane damage, and blurred and dissolved mitochondrial cristae (Figure 3G). Considered together, the above results demonstrated that PNPLA3-I148M causes abnormalities in mitochondrial morphology.

### 2.5. Effect of PNPLA3 on the Expression of Mitochondrial Proteins

We examined protein and gene expression levels associated with mitochondrial function. Compared with PNPLA3-I148I, PA-retinol incubation significantly decreased the gene expression levels of *PTEN-induced putative kinase 1 (PINK1)* (*p* < 0.001), *BCL2 interacting protein 3 (BNIP3)* (*p* < 0.05), *mitochondrial dynamics-related genes (MFN1)* (*p* < 0.05), *MFN2* (*p* < 0.05), mitochondrial *biosynthesis related genes-sirtuin 1* (SIRT1) (*p* < 0.05), *AMP-activated protein kinase (AMPK)* (*p* < 0.01), *superoxide dismutase 1 (SOD1)* (*p* < 0.05), and mitochondrial *transcriptional factor A (TFAM)* (*p* < 0.001) in PNPLA3-I148M cells, while *mitochondrial fission protein 1* (FIS1) (*p* < 0.001), *dynamin-related protein 1* (DRP1) (*p* < 0.01), and *Optic atrophy type 1 (OPA1)* (*p* < 0.01) expression was increased (Figure 4A). After PA-retinol depletion, most genes in PNPLA3-I148M cells maintained the same expression changes compared to incubation conditions, except for *FIS1* and *DRP1* which did not differ between PNPLA3-I148I and -I148M (Figure 4B). Interestingly, elevated *uncoupling protein 3 (UCP3)* (*p* < 0.05) and reduced *nuclear factor erythroid 2-related factor 2 (NRF2)* (*p* < 0.01) gene expression in PNPLA3-I148M was found only after PA-retinol depletion compared to PNPLA3-I148I cells (Figure 4B). In terms of protein levels, *SOD-1* (*p* < 0.05) and *MFN2* (*p* < 0.01), but not *DRP1*, were significantly decreased in the PNPLA3-I148M group compared to other groups under depletion conditions (Figure 4C–G). These findings were consistent with the gene expression results and suggested that PNPLA3-I148M disrupts the expression patterns of functional proteins associated with mitochondria.

### 2.6. PNPLA3-I148M Inhibits Mitochondrial Respiration in LX-2 Cells

To better understand the effects of PNPLA3-I148M on mitochondrial physiology, we analyzed the oxygen consumption rate (OCR) of these living LX-2 cell lines using the SeaHorse XF96 analyzer. PNPLA3-I148I cells were able to enhance OCR after FCCP treatment, while EV and PNPLA3-I148M were not significantly altered (Figure 5A,B). Moreover, EV and PNPLA3-I148M showed lower maximal respiration (*p* < 0.001), proton leak (*p* < 0.001), non-mitochondrial respiration (*p* < 0.001), basal mitochondrial oxygen consumption (*p* < 0.001), and ATP production (*p* < 0.001) compared to PNPLA3-I148I cells (Figure 5C–G), suggesting that PNPLA3-I148M can promote a cellular bioenergetics imbalance by significantly inhibiting all mitochondrial parameters, which may contribute to the activation of LX-2 cells and the progression of fibrosis.

## 3. Discussion

In the last decade, an increasing number of events have demonstrated that PNPLA3 rs738409 polymorphism is closely associated with hepatic fat accumulation, ALD, NASH, cirrhosis, and HCC [15,17,25,26,27,28]. It has been further reported that the PNPLA3-I148M variant promotes the progression of fibrosis by modulating retinol metabolism and upregulating the pro-inflammatory cytokines JNK and AP-1 in HSCs or LX-2 cells upon different types of nutritional status, respectively [20,21,22]. However, the molecular mechanisms linking PNPLA3-driven dysregulation of lipid metabolism to the occurrence of fibrosis remain largely unknown. In the current study, we detected the impact of PNPLA3-I148M on cholesterol metabolism and mitochondrial function in LX-2 cells. The main findings of this study are as follows: (i) overexpression of PNPLA3-I148M resulted in lipid droplet accumulation and a fibrotic phenotype in LX-2 cells stimulated by PA-retinol, which was even more pronounced under PA-retinol-depleted conditions accompanied by higher gene expression levels of *Col1a1*, *Col3a1*, *MMP2*, *TIMP1*, and *TIMP2*, and an increased trend of *AP-1* and *MLC-2* expression; (ii) PNPLA3-I148M blocks the cholesterol efflux pathway by reducing the expression of the ABCG1 protein, resulting in the accumulation of FC in LX-2 cells; (iii) we demonstrate, for the first time, that PNPLA3-I148M leads to mitochondrial dysfunction characterized by increased reactive oxygen species, decreased ATP product and ΔΨm, impaired mitochondrial morphology, and the inhibition of mitochondrial respiration in LX-2 cells after PA-retinol removal, which contribute to the development of fibrosis (Figure 6).

It has been reported that HSCs, hepatocytes, and LX-2 cells have high expression of the endogenous PNPLA3 protein [20]. Moreover, PA and retinol were also used to induce a fibrotic phenotype of LX-2 cells carrying the PNPLA3-I148M protein [20,21,22]. On this basis, we generated stable LX-2 cell lines overexpressing PNPLA3-I148I or -I148M and observed consistent results with previous studies, such as the localization of the PNPLA3 protein on the surface of LD membranes, and a high LD level and fibrosis-related gene expression in PNPLA3-I148M cells [20,21,29]. The above results indicated that our stable LX-2 cell lines have a similar PNPLA3-I148M-related phenotype compared to other studies.

Emerging evidence in recent years suggests that FC accumulation plays a critical role in the progression of NASH and liver fibrosis [12,30,31]. The accumulation of FC in mouse HSCs has been shown to lead to more severe fibrosis through increased Toll-like receptor 4 protein levels and the subsequent sensitization of the cells to TGF-β [32]. Furuhashi et al. [33] found that intracellular FC aggregation can trigger a vicious cycle of HSC activation, which is independent of serum cholesterol levels in vivo. Moreover, a deficiency of the cholesterol metabolism-related gene *Acyl-coenzyme A: cholesterol acyltransferase (ACAT1)* or *Niemann-Pick type C2 (NPC2)* increases the levels of FC in HSCs, thereby exacerbating the progression of liver fibrosis [34,35]. In this study, we found that PNPLA3-I148M functioned to promote FC aggregation in LX-2 cells compared with EV and PNPLA3-I148I cells. In particular, the intracellular levels of FC in PNPLA3-I148M were 29.1% higher than after incubation conditions after the removal of PA-Retinol (Figure 5C). The results indicated that PNPLA3-I148M could lead to the intracellular retention of FC, which was exacerbated after the removal of the stimulus.

Cholesterol is an essential structural component of cell membranes and an important precursor for steroid hormones, bile acids, and vitamin D synthesis. The maintenance of intracellular cholesterol homeostasis depends on the balance of the cholesterol absorption, endogenous synthesis, transport, esterification, and efflux pathway. Bruschi et al. [22] reported that PNPLA3-I148M inhibited LXR signaling and promoted the accumulation of FC in HSCs isolated from donor livers unsuitable for transplantation and stimulation with the LXR agonist PPARγ agonist; this was accompanied by the reduced expression of genes or proteins related to cholesterol efflux (ABCA1), synthesis (HMGCR), and uptake (LDLR). However, we observed no significant effect of PA-retinol incubation or depletion on protein expression of HMGCR, LDLR, and SREBP2 between the PNPLA3-I148I and -I148M groups, which is contrary to previous studies. This discrepancy may be due to different cell sources and stimulation conditions. Since both *LDLR* and *HMGCR* are downstream target genes of *SREBP2*, our study showed that PNPLA3-I148M has no effect on cholesterol biosynthesis and uptake in the LX-2 cell line.

Moreover, the expression of ACAT1 protein, a rate-limiting enzyme that catalyzes the formation of cholesteryl ester from FC [34], was significantly increased in the PNPLA3-I148M group only under PA-retinol depletion. The results indicated that PNPLA3-I148M promoted the overaccumulation of cholesterol esters, which were stored in lipid droplets and further exacerbated the lipid overload state in LX-2 cells. In terms of cholesterol efflux, we found that ABCA1 protein levels were significantly higher in the PNPLA3-I148M group under both stimulation conditions, whereas ABCG1 was obviously decreased in PNPLA3-I148M cells only under PA-retinol depletion conditions. ABCA1 has been shown to mediate the initial intracellular transport of FC to apolipoprotein A-I (apoA-I) for the formation of nascent high-density lipoprotein (HDL) particles; then, ABCG1 promotes HDL maturation and ultimate excretion by facilitating cholesterol efflux to the nascent HDL [36,37]. However, the reduced protein expression levels of ABCG1 in the current study indicated that PNPLA3-I148M contributes to the retention of intracellular FC in LX-2 cells under depletion conditions, which might feedback induce the expression of ATAC1 and ABCA1 proteins to produce more esterified forms of cholesterol or nascent HDL.

Mitochondria are important cholesterol-poor cell organelles with double membrane-bounded structures that generate the energy required for basic cellular functions, such as metabolism, ROS, apoptosis, etc. [38,39]. It has been reported that mitochondrial cholesterol overloading is closely related to various pathological conditions of liver disease, including steatosis, fibrosis, and HCC [38,40]. In our study, PNPLA3-I148M LX-2 cells were shown to have reduced ATP and membrane potential under PA-retinol depletion conditions, accompanied by high levels of ROS production. Mitochondrial membrane potential is an essential component for ATP production achieved by supplying the electrochemical potential of hydrogen ions; it also plays an important role in regulating the oxidative phosphorylation state of mitochondria [41]. ROS have also been reported to promote the development of hepatic fibrosis by directly regulating the expression and metabolism of fibrosis-associated matrix proteins [42]. The loss of mitochondrial membrane potential, reduced ATP generation, and oxidative stress caused by elevated intracellular ROS levels have all become hallmarks of mitochondrial dysfunction in NASH and advanced fibrosis [43,44,45]. Taken together, our results suggested that PNPLA3-I148M-induced FC aggregation promotes a further increase in mitochondrial-cholesterol content, which ultimately leads to mitochondrial dysfunction in LX-2 cells.

Mitochondrial cholesterol overload has been reported to alter the membrane’s physical properties, organization, permeability, and resident protein function, resulting in the weakening of the mitochondrial antioxidant defense [46,47]. To elucidate the effects of the PNPLA3-I148M-induced intracellular aggregation of FC on the mitochondrial structure, we analyzed the changes in mitochondrial morphology. PNPLA3-I148M LX-2 cells display an obvious mitochondrial damage phenotype, characterized by the disruption of the outer membrane, blurred and dissolved cristae, and dark matrix when compared to PNPLA3-I148I and EV. The cristae, containing mitochondrial DNA (mtDNA) with a protein-rich matrix, are sub-compartments of the inner membrane and the oxidative phosphorylation site of the mitochondria, which play a critical role in maintaining mitochondrial membrane potential (ΔΨm) and ATP production [48]. Damage to the structure of cristae has been closely associated with the pathophysiology of a variety of diseases [48]. The above results suggested that PNPLA3-I148M promotes structural damage of mitochondria, which may be due to the overload of FC in LX-2 cells.

We then analyzed the expression levels of proteins and genes involved in mitochondrial function. The results showed that the expression of genes involved in mitosis *(PINK*, *BNIP3)*, mitochondrial fusion *(MFN1*, *MFN2*, *SRIT1)*, mitochondrial biogenesis *(AMPK)*, and antioxidant enzymes *(SOD-1)* was significantly altered in the PNPLA3-I148M group compared to the PNPLA3-I148I group under PA-retinol depletion conditions, along with significantly increased protein levels of SOD-1 and MFN2. SOD1 is a primary antioxidative enzyme that plays a key role in the intracellular antioxidant defense system [49]. SOD-1 knockout mice were shown to cause liver fibrosis by promoting MMP (MMP 2 and 9) and TIMP (TIMP1) protein expression; collagen accumulated in the liver [50,51]. Meanwhile, the liver-specific ablation of MFN2 in mice has been reported to induce inflammation, fibrosis, and even liver cancer, whereas the overexpression of MFN2 was able to ameliorate CCl4-induced liver fibrosis by inhibiting the TGF-β1/Smad signaling pathway and downregulating collagen levels, such as type I, type III, and type IV collagen [52,53]. In combination with our findings, intracellular cholesterol aggregation leads to mitochondrial damage and also further promotes the altered expression of mitochondrial-related proteins, which is an important mechanism for PNPLA3-I148M to induce LX-2 cell activation and the initiation of fibrosis.

Finally, we measured the oxygen consumption rate (OCR) of LX-2 living cells to elucidate the effect of PNPLA3-I148M on mitochondrial physiology. PNPLA3-I148I cells were able to enhance OCR after the FCCP treatment, while PNPLA3-I148M was not significantly altered, which may reflect the effect of PNPLA3-I148M on spare respiratory capacity. Moreover, PNPLA3-I148M cells showed lower basal mitochondrial oxygen consumption, non-mitochondrial respiration, proton leak, maximal respiration, and ATP production compared to PNPLA3-I148I, suggesting that PNPLA3-I148M promote a cellular bioenergetics imbalance by inhibiting mitochondrial parameters, leading to mitochondrial dysfunction and the subsequent development of fibrosis.

The main limitation of our study is that the effect of PNPLA3 on cholesterol metabolism and mitochondrial function was only explored in the hepatic stellate cell line LX-2; further in vivo studies are needed to support this conclusion.

In conclusion, our study revealed that PNPLA3-I148M promotes FC accumulation in LX-2 cells by reducing ABCG1 protein expression and inhibiting cholesterol efflux. Intracellular cholesterol retention further disrupts the mitochondrial structure and function, leading to the decreased expression of mitochondria-related proteins, such as SOD-1 and MFN2, ultimately promoting the activation of LX-2 cells and the development of a fibrotic phenotype. These findings provide new evidence for the regulatory mechanisms of PNPLA3-I148M on lipid metabolism, mitochondrial dysfunction, and liver fibrosis, which will help to improve the understanding of the biological functions of PNPLA3 mutants and facilitate the development of therapeutic strategies for patients with NAFLD carrying the I148M allele.

## 4. Materials and Methods

### 4.1. Cell Culture

The immortalized human stellate cell line LX-2 was cultured in Dulbecco’s Modified Eagle Medium (DMEM) (Gibco, Billings, MT, USA) and supplemented with 2% fetal bovine serum (Gibco, Billings, MT, USA) and 1% penicillin/streptomycin solution (Gibco, Billings, MT, USA) at 37 °C and 5% carbon dioxide (CO_2_), according to previous studies [54]. Subsequent passages were performed every 3 days, and cells were digested using trypsin (0.05% trypsin/0.53 mM EDTA) (Gibco, Billings, MT, USA) and inoculated at a 1:3 ratio.

### 4.2. Generating Stable Cell Line

The pLV[Exp]-Puro-CMV-Flag vector containing an empty vector (EV) and human PNPLA3 I148I (wild type) or-I148M (mutant) cDNA was kindly provided by Prof. Wanqing Liu (Wayne State University, Detroit, MI, USA) and transfected into 293T cells (2.5 × 10^5^ cell in 6 well plates) with the plasmids pMDLg, pVSVg, and RSV-Rev using Liptofectamine 3000 reagent (ThermoFisher, Waltham, MA, USA) for lentiviral packaging. The medium was replaced with fresh medium overnight after transfection and was incubated at 37 °C with 5% CO_2_ for an additional 48 h. The supernatant containing lentiviral particles was collected and centrifugation was conducted at 3000× *g* for 5 min at 4 °C; then, it was filtered through a 0.45-μm filtration (Sigma, St. Louis, MO, USA). Viral supernatant was used to infect the LX-2 cells (1 × 10^6^) with 8 μg/mL polybrene for 8 h at 37 °C with 5% CO_2_ before being replaced with fresh medium. Finally, LX-2 cells were treated with 0.2 μg/mL puromycin for 7 days to obtain a stable cell line.

For LX-2 cell activation, two different conditions were used, according to previous reports [20]. Incubation conditions: LX-2 cells were incubated with 100 nM BSA- (fatty-acid-free) conjugated palmitic acid (PA) (Sigma, St. Louis, MO, USA) and 10 μM retinol (Sigma, St. Louis, MO, USA) for 24 h at 37 °C and 5% CO_2_ with regular medium. Depletion condition: LX-2 cells were incubated with PA and retinol for 24 h and then replaced with a regular medium for an additional 24 h. Images of LX-2 cells under conditions of untreated, PA-retinol incubation and PA-retinol depletion stimulation are provided in the Appendix A.

### 4.3. Lipid Analysis

For the isolation of total cholesterol (TC), 1 × 10^6^/mL LX-2 cells were lysed by 0.1% NP40 lysis buffer and then lipids were extracted with chloroform:isopropanol (7:11) in a micro-homogenizer. The sample was centrifuged at 13,000× *g* for 10 min and then air-dried at 50 °C to remove residual organic solvents. The pellets were dissolved in the Cholesterol Assay Buffer (Sigma, St. Louis, MO, USA) and vortexed until the mixture was homogenous. For the isolation of triglycerides (TG), 1 × 10^6^/mL LX-2 cells were lysed by IGEPAL CA-630 (Beyotime, Shanghai, China). The sample was slowly heated to 100 °C in a metal bath for 3 min and then cooled to room temperature (RT). Centrifugation took place at 16,000× *g* for 2 min to collect the supernatant. The amounts of TG, TC, and FC were measured using a cholesterol test kit (Sigma, St. Louis, MO, USA) and triglyceride detection kit (Sigma, St. Louis, MO, USA), respectively, according to the manufacturer’s instructions. Protein concentrations were quantified by the BCA protein quantitative analysis kit (ThermoFisher, Waltham, MA, USA).

### 4.4. BODIPY Staining

The LX-2 cells (5 × 10^5^/mL) were seeded in a 35 mm cell imaging dish with a glass bottom (Cellvis, Mountain View, CA, USA). The cells were incubated with 100 μM palmitic acid and 10 μM retinol for 24 h at 37 °C with 5% CO_2_ and then fixed with 4% paraformaldehyde (Beyotime, Shanghai, China) for 15 min at RT. In addition, 2 μM BODIPY (493/503) staining solution in PBS was added into the cells and incubated for 1 h at 37 °C. The cells were mounted with an anti-fading mounting medium with DAPI (Sigma, St. Louis, MO, USA) and fluorescence images were obtained with a confocal microscope (TCS SP8, Leica, Weztlar, Germany).

### 4.5. Filipin III Staining

The LX-2 stable cells were fixed with 4% paraformaldehyde after stimulation, and then paraformaldehyde was quenched by incubation with 1.5 mg/mL glycine (Beyotime, Shanghai, China) in PBS for 10 min at RT. For free cholesterol staining, 0.05 mg/mL Filipin III (Sigma, St. Louis, MO, USA) and 200 μL DiO cell membrane green fluorescence staining solution (Beyotime, Shanghai, China) were added to LX-2 cells, followed by incubation for 2 h at RT. Cells were mounted with a DAPI-free anti-fading mounting medium (Solarbio, Beijing, China) and covered with cover slides. The fluorescence of the cells was observed using a confocal microscope (TCS SP8, Leica, Gremany).

### 4.6. Co-Localization Analysis

LX-2 cells (1 × 10^5^/mL) were seeded into a 35 mm cell imaging dish with a glass bottom. After PA-retinol incubation or depletion, the cells were fixed in 4% paraformaldehyde for 15 min and permeabilized with PBS containing 0.3% TritonX-100 for 15 min at RT. Cells were blocked with 5% bovine serum albumin (BSA) in PBS for 1 h, then incubated with mouse anti-Flag polyclonal affinity antibody (1:200, Thermofisher, Waltham, MA, USA) at 4 °C overnight, followed by an Alexa Fluor^®^ 647 conjugate goat anti-Mouse IgG (H + L) cross-adsorbed secondary antibody at RT for 2 h. After being washed by PBS containing 1% Tween 20, the cells were incubated with 2 μM BODIPY for 2 h at RT. Finally, the cells were mounted with DAPI containing anti-fade mounting medium and fluorescence images were acquired with a confocal microscope (TCS SP8, Leica, Germany).

### 4.7. ATP Measurements

The LX-2 cells (5 × 10^5^) were seeded into each well of a 6-well plate. After PA-retinol incubation or depletion, the cells were lysed by adding 200 μL ATP detection lysis buffer (Beyotime, Shanghai, China) and they were centrifuged at 4 °C and 12,000× *g* for 5 min. After transferring 100 μL of ATP detection buffer (Beyotime, Shanghai, China) to a 96-well plate, incubation took place for 5 min at RT and 20 μL cell lysate supernatant was added. Relative light units (RLU) were detected by a chemiluminescence instrument (Cytation3, Biotek, Winooski, Vt, USA). Protein concentration was determined with the BCA protein quantitative analysis kit (ThermoFisher, Waltham, MA, USA).

### 4.8. Analysis of Reactive Oxygen Species

Reactive oxygen species (ROS) were measured by a ROS detection kit (Beyotime, Shanghai, China). The cells were incubated with 1 mL serum-free medium containing 10 μm/L of the fluorescent probe DCFH-DA (Beyotime, Shanghai, China) at 37 °C for 20 min. The cells were washed 3 times with a serum-free cell culture medium to completely remove residual DCFH-DA. Fluorescence was measured at the excitation/emission (ex/em) wavelength of 488/525 nm.

### 4.9. Measurement of Mitochondrial Transmembrane Potential (ΔΨm)

The ΔΨm was determined using an enhanced mitochondrial membrane potential detection kit (JC-1) (Beyotime, Shanghai, China). In short, 5 × 10^5^ cells were resuspended in 0.5 mL cell culture medium; then, 0.5 mL JC-1 staining medium with JC-1 fluorescent probe (Beyotime, Shanghai, China) was added and mixed. After incubation at 37 °C for 20 min, the cells were harvested by centrifugation at 600× *g* for 3 min at 4 °C and then washed twice with JC-1 staining buffer (Beyotime, Shanghai, China). The cell pellets were resuspended in 0.5 mL JC-1 staining buffer. The fluorescence was read at the ex/em wavelength of 485/590 nm.

### 4.10. Oxygen Consumption Rate Measurements

LX-2 cells (2 × 10^5^) were seeded in the hippocampal XF96 cell culture plate (SeaHorseBioscience, Palo Alto, CA, USA) and placed in a cell culture incubator at 37 °C, 5% CO_2_ overnight. The culture medium was replaced with XF Base Medium (pH 7.4) supplemented with 1.0 M Glucose Solution, 100 mM Pyruvate Solution, and 200 mM Glutamine Solution (SeaHorseBioscience, Palo Alto, CA, USA) after stimulation; it was then incubated in a CO_2_-free cell incubator at 37 °C for 60 min. OCR was measured using the Seahorse XF cell mito stress test kit (SeaHorseBioscience, Palo Alto, CA, USA), according to the manufacturer’s instructions. Oligomycin (1.5 μM), FCCP (0.5 μM), and Rot/AA (0.5 μM) were added to each well; then, the plate was transferred to a SeaHorseXF96 analyzer (SeaHorseBioscience, Palo Alto, CA, USA) for analysis. OCR was normalized according to the number of cells.

### 4.11. Mitochondrial Ultrastructure

The cells were collected by centrifugation at 3000 rpm for 2 min and resuspended in 2.5% glutaraldehyde fixed solution for 30 min at RT. The cells were washed 3 times with 0.1 M PB buffer (pH 7.4) and post-fixed in 1% buffered OsO_4_ for 2 h at RT in the dark. The process of dehydration, resin infiltration, and embedding was performed by Servicebio company, according to a standard operating procedure. The ultrastructures of the mitochondria were observed by a HT-7800 transmission electron microscope (HITACHI, Tokyo, Japan).

### 4.12. Real-Time PCR

Total RNA was extracted from cells by the Trizol (Sigma, St. Louis, MO, USA) method, following the manufacturer’s instructions. The extracted RNA was reverse transcribed into cDNA by the HiScript II 1st strand cDNA synthesis kit (Vazyme, Nanjing, China). The SYBR Green Master Mix (Thermofisher, Waltham, MA, USA) and ViiA fluorescence 7 real-time fluorescence quantitative PCR reaction system (Applied Biosystems, Foster City, CA, USA) were used for quantitative RT-PCR. All primers were synthesized from Sangon Biotech (Shanghai, China) and listed in the Appendix A. The cycle conditions of the polymerase chain reaction are as follows: 50 °C for 2 min, 95 °C for 2 min, and then 40 cycles at 95 °C for 15 s and 60 °C for 1 min. The threshold cycle (2^−ΔΔCT^) method was used to calculate the folding change of relative gene expression.

### 4.13. Western Blot

The cells were lysed in RIPA lysis buffer (Beyotime, Shanghai, China) with 1 mM PMSF and 10 ug/mL complete protease inhibitor mixture (Roche, Basel, Switzerland). The protein concentration was measured by a BCA protein quantitative analysis kit (ThermoFisher, Waltham, MA, USA). The protein samples were separated by 10% SDS-PAGE gel electrophoresis and then transferred to a PVDF membrane (Merck, Kenilworth, NJ, USA). The membranes were blocked with 5% non-fat milk in Tris-buffered saline containing 0.1% Tween-20 (TBST) and incubated with primary antibodies (Appendix A) at 4 °C overnight. After washing in TBST, the membranes were further incubated with a horseradish peroxidase-labeled goat anti-rabbit antibody (Cell Signaling Technology, Danvers, MA, USA). The protein bands were captured using the ChemiDocTM MP Imaging System (Bio-Rad, Hercules, CA, USA) and quantified with Image J Software (Version 1.51K).

### 4.14. Statistical Analysis

The data were expressed as mean ± standard deviation. Statistical analysis of data was performed using GraphPad Prism version 8.0.2 (GraphPad Software, San Diego, CA, USA). The Student’s *t*-test was used for comparison between two groups, and a one-way analysis of variance was used for comparison between multiple groups. Three independent verifications were carried out for each experiment. A *p*-value of less than 0.05 was considered statistically significant.

## Figures and Tables

**Figure 1 ijms-24-09681-f001:**
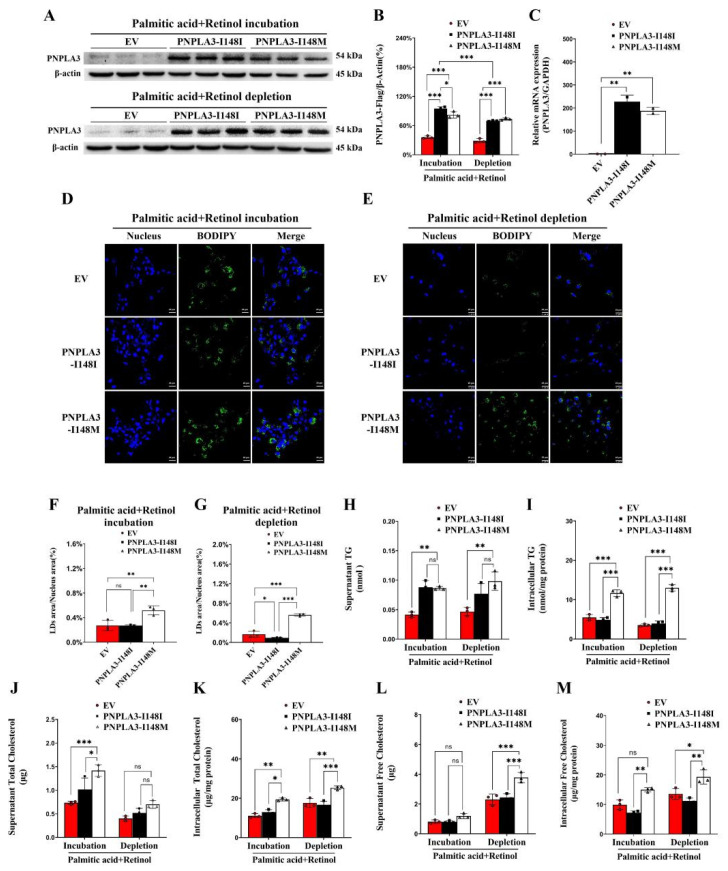
Overexpression of PNPLA3-I148M promotes lipid accumulation in LX-2 cells carrying EV, PNPLA3-I148I, or-I148M under PA-retinol incubation and depletion conditions. (**A**) Expression of PNPLA3-I148I or-I148M protein. (**B**) Quantification of PNPLA3 protein. (**C**) Gene expression of PNPLA3-I148I or-I148M in LX-2 cells. Distribution of lipid droplets in LX-2 cells stably overexpressing EV, PNPLA3-I148I, and -I148M under PA-retinol incubation (**D**) and depletion (**E**) conditions. Blue: nucleus, green: lipid droplets. Scale bars: 25 μm, 400× magnification. (**F**,**G**) Quantification of lipid droplet area in LX-2 cells. (**H**–**M**) The supernatant and intracellular levels of TG, TC, and FC in each group under PA-retinol incubation or depletion conditions. n = 3, * *p* < 0.05, ** *p* < 0.01, *** *p* < 0.001. EV, empty vector; TG, triglyceride; TC, total cholesterol; FC, free cholesterol; PA: palmitic acid.

**Figure 2 ijms-24-09681-f002:**
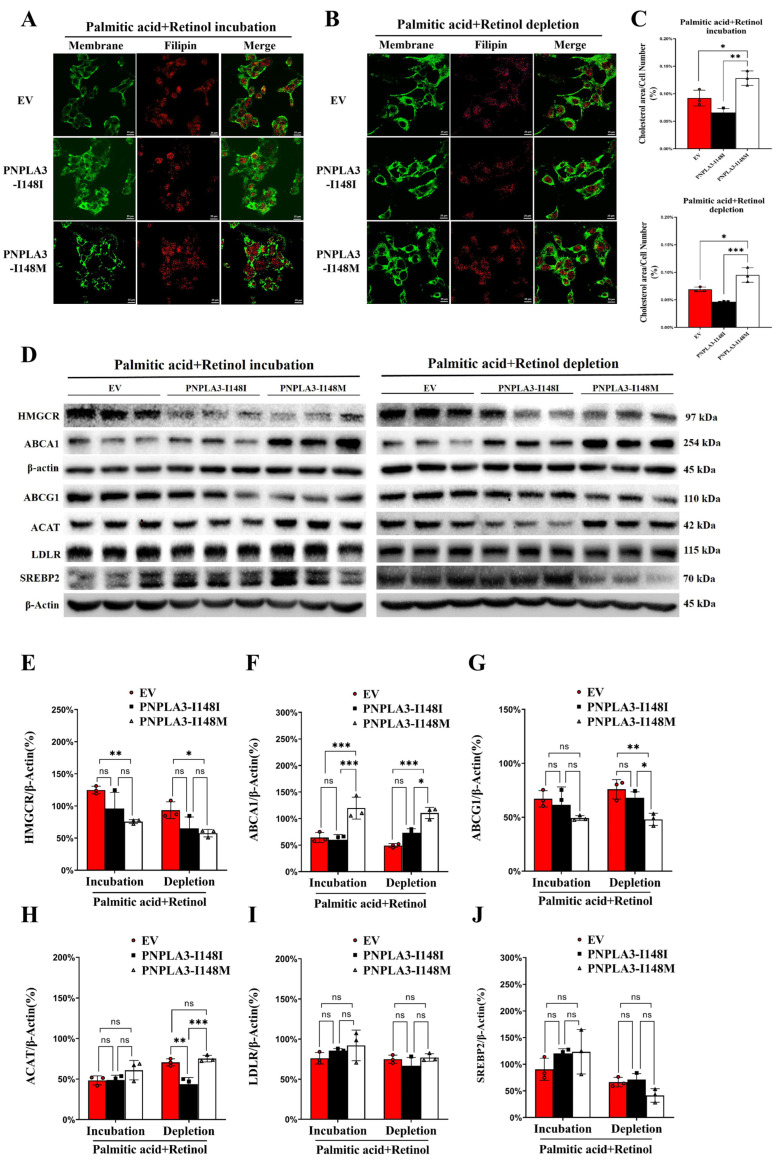
Distribution of free cholesterol and expression of cholesterol metabolism-related protein in LX-2 cells. (**A**,**B**) Distribution of free cholesterol in each group. Green: cell membrane, red: free cholesterol. Scale bars: 25 μm, 400× magnification. (**C**) Quantification of free cholesterol. (**D**) Expression of HMGCR, ABCA1, ABCG1, ACAT, LDLR, and SREBP2 proteins. (**E**–**J**) Quantification data of the identified proteins. Three independent experiments were performed. n = 3, * *p* < 0.05, ** *p* < 0.01, *** *p* < 0.001.

**Figure 3 ijms-24-09681-f003:**
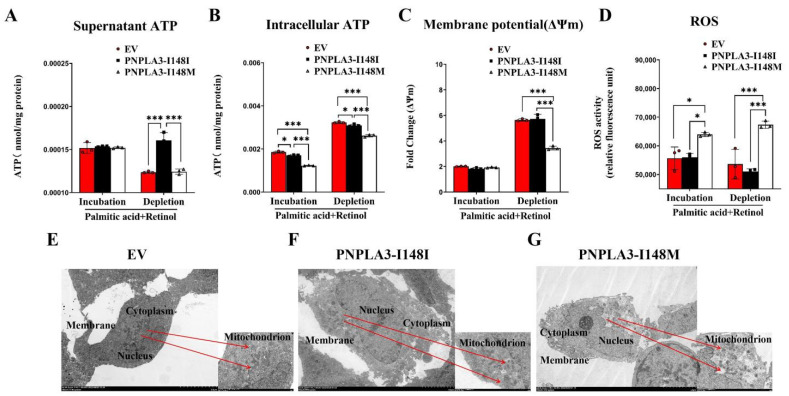
Analysis of mitochondrial function and morphology in LX-2 cells under PA-retinol incubation or depletion groups. (**A**,**B**) Measurement of the supernatant and intracellular ATP levels. (**C**) Mitochondrial membrane potential analysis. (**D**) Detection of mitochondrial ROS levels. (**E**–**G**) Mitochondrial morphology analysis in each group. Scale bars: 5.0 μm, 250× magnification. n = 3, * *p* < 0.05, *** *p* < 0.001.

**Figure 4 ijms-24-09681-f004:**
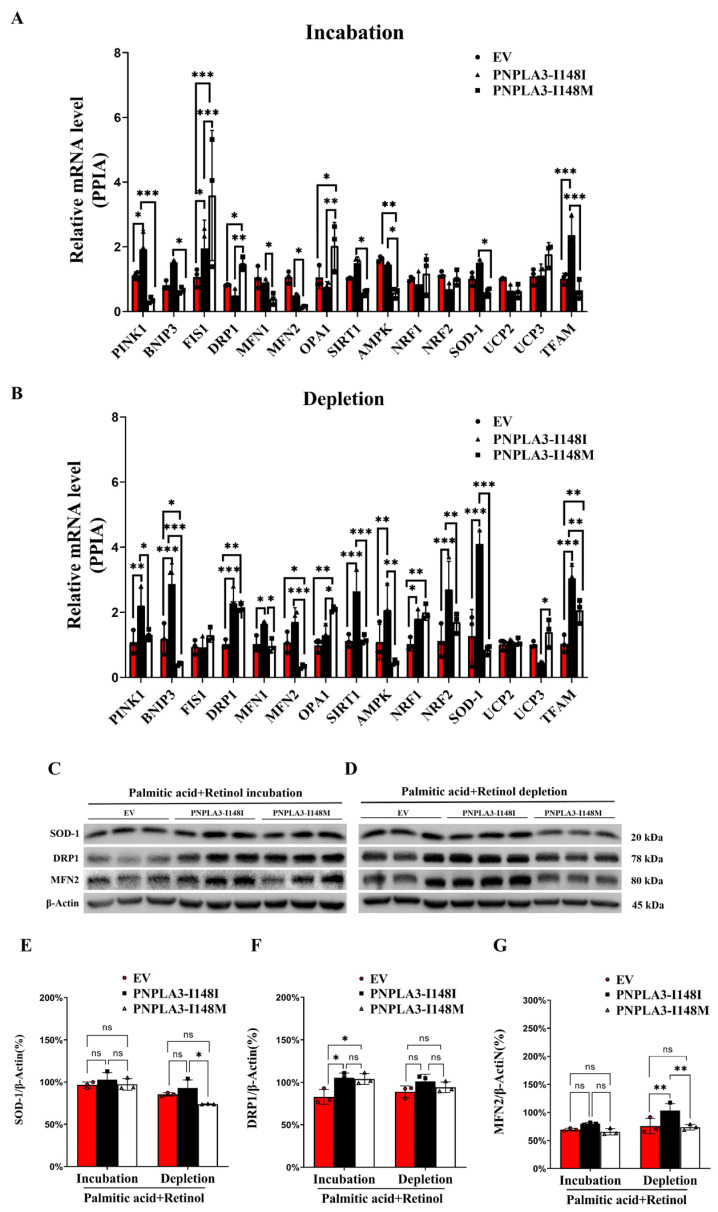
The expression of genes and proteins involved in mitochondrial function. (**A**,**B**) mRNA expression of genes related to mitochondrial function in each group under PA-retinol incubation and depletion conditions. (**C**,**D**) Protein expression of SOD-1, DRP1 and MFN2 in each group. (**E**–**G**) Quantification of SOD-1, DRP1 and MFN2 protein. n = 3, * *p* < 0.05, ** *p* < 0.01, *** *p* < 0.001.

**Figure 5 ijms-24-09681-f005:**
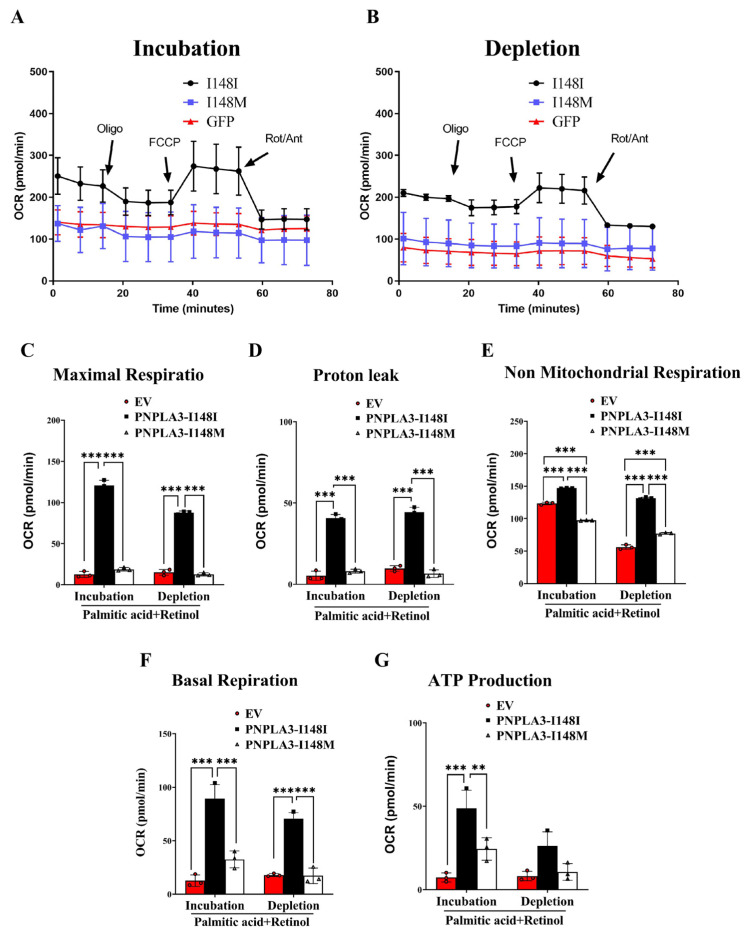
Seahorse analysis of cellular respiration. (**A**,**B**) Measurement of oxygen consumption rate (OCR) in LX-2 cells carrying EV, human PNPLA3-I148I, or-I148M under PA-retinol incubation and depletion conditions. n = 8, Red: EV, Black: PNPLA3-I148I, Blue: PNPLA3-I148M. Quantification of maximal respiration (**C**), proton leak (**D**), non-mitochondrial oxygen consumption (**E**), basal respiration (**F**), and ATP product (**G**) in each group. Data are expressed as mean ± SEM, ** *p* < 0.01, *** *p* < 0.001.

**Figure 6 ijms-24-09681-f006:**
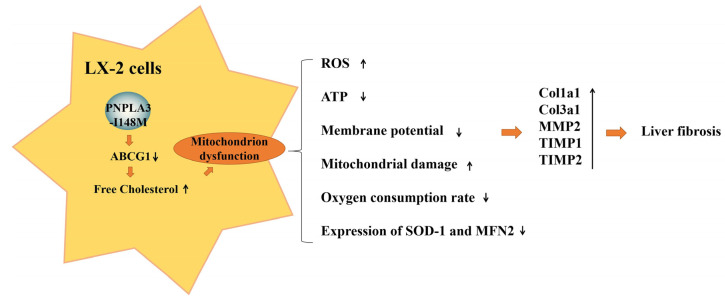
Schematic diagram of the signaling pathway of PNPLA3-I148M-induced activation of LX-2 cells. In the presence of PNPLA3-I148M, the expression of the cholesterol efflux protein ABCG1 was significantly reduced, leading to the accumulation of FC in LX-2 cells and the disruption of mitochondrial cholesterol homeostasis. The diminished mitochondrial function and structural damage further promoted the activation of LX-2 cells and increased the expression of fibrosis-related factors, which may be a key mechanism of PNPLA3-I148M leading to the progression of liver fibrosis. Upward arrows indicate upregulation and downward arrows indicate downregulation.

## Data Availability

The data presented in this study are available upon request from the corresponding author.

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
