# Peer review of "PNPLA3-I148M Variant Promotes the Progression of Liver Fibrosis by Inducing Mitochondrial Dysfunction"

_ijms, 2023, doi:10.3390/ijms24119681_

Round 1
Reviewer 1 Report (Previous Reviewer 2)
The authors have been responsive to the concerns raised in the prior evaluation of the manuscript.
Author Response
Response:We appreciate this reviewer’s kind review and comments, which will further improve our work.
Reviewer 2 Report (New Reviewer)
The authors investigated the effect of PNPLA3-I148M on the activation of the hepatic stellate cell line LX -2 and the progression of liver fibrosis in these cells. They state that PNPLA3-I148M significantly promoted intracellular aggregation of free cholesterol in LX -2 cells through decreased expression of the cholesterol efflux protein ABCG1 and subsequently induced mitochondrial dysfunction, which was characterised by decreased ATP production and mitochondrial membrane potential, increased ROS levels, mitochondrial structural damage, altered oxygen consumption rate and decreased expression of mitochondrial function-related proteins. Partially similar data have been published with primary HSC:s and LX2 cells, as the authors also mention. The relationship between the effects and the change in ABCG1 expression is interesting and the aspects focusing on the effects of the mutant on mitochondrial function using several different endpoints. Overall, I think the study adds to the knowledge of the role of PNPLA3-I148M in NASH.
The data are shown only in a transformed cell line that mimics stellate cell function in many aspects. Conclusions using primary HSCs from different donors or more integrated liver systems for studies of NASH such as liver cells in physiologically relevant microfluidic plates or in liver spheroids would improve the conclusions regarding the clinical significance of the results.
Special comments
The text size in all figures is not compatible with efficient reading. This is particularly true for Fig. 1 B & C and F-M, Fig. 2 E-J, Fig. 3 E-G, Fig. 4 E-G. Fig. 5 C-G and most importantly to Fig 6. The information is lost for readers of the printed version of the manuscript.
Fig. 2
In A (palmitin+retinol incubations), the data from the figure are not convincing that the levels of FC were significantly higher in PNPLA3-I148M cells. How many independent experiments were performed? How many slides were quantified per experiment? This is not clear from the legend.
The main point raised by the authors is the decrease in ABCG1 by PNPLA3-I148M. However, this appears to be due to the expression levels in three lanes. The reproduce a Western blotting experiment using the same samples is often difficult. What does this actually mean? Mean values of three independent experiments? How was the intraexperimental variation? Was this difference reproducible?
What is actually stained for in the images of the membranes in A and B? What are cholesterol-related membrane proteins?
Fig. 3
Significant depletion of intracellular ATP is shown in both FFA treated and FFA depleted cells of variant I148M. How can this be explained?
Using Insta-text for language improvement is recommended!
Author Response
Please see the attachment.

This manuscript is a resubmission of an earlier submission. The following is a list of the peer review reports and author responses from that submission.
Round 1
Reviewer 1 Report
I will highly recomment to the authors to find the nobelty in their research. Everything pointed out here has been published 4 years ago!
Reviewer 2 Report
This is an interesting paper that presents a novel line of investigation into potential mechanisms by which the common rs738409 variant might promote hepatic fibrosis. There are no major concerns with the experimental design or interpretation of the findings. The one recommendation I offer is the inclusion of images of the LX-2 cells under the conditions utilized (as a supplemental file).
While a number of knockdown/overexpression experiments might be proposed and a primary cell line suggested for replication, this reviewer recognizes that those experiments lie outside the scope of this study.
There are some instances of typographical errors or incorrect grammar that need to be addressed to improve the flow and quality of the manuscript.